# Scope 3 emission estimation using large language models

Ayush Jain*
Manikandan Padmanaban*
ayush.jain@ibm.com
manipadm@in.ibm.com
IBM Research Labs, India

Jagabondhu Hazra
Shantanu Godbole
jahazra1@in.ibm.com
shantanugodbole@in.ibm.com
IBM Research Labs, India

Kommy Weldemariam
kommy@ibm.com
IBM Research Labs, USA

## ABSTRACT

Large enterprises face a crucial imperative to achieve the Sustainable Development Goals (SDGs), especially goal 13, which focuses on combating climate change and its impacts. To mitigate the effects of climate change, reducing enterprise Scope 3 (supply chain emissions) is vital, as it accounts for more than 90% of total emission inventories. However, tracking Scope 3 emissions proves challenging, as data must be collected from thousands of upstream and downstream suppliers. To address the above mentioned challenges, we propose a *first-of-a-kind* framework that uses domain-adapted NLP foundation models to estimate Scope 3 emissions, by utilizing financial transactions as a proxy for purchased goods and services. We compared the performance of the proposed framework with the state-of-art text classification models such as TF-IDF, word2Vec, and Zero shot learning. Our results show that the domain-adapted foundation model outperforms state-of-the-art text mining techniques and performs as well as a subject matter expert (SME). The proposed framework could accelerate the Scope 3 estimation at Enterprise scale and will help to take appropriate climate actions to achieve SDG 13.

## KEYWORDS

scope 3 emission, large language models, SDG goals 13

**ACM Reference Format:**
Ayush Jain, Manikandan Padmanaban, Jagabondhu Hazra, Shantanu Godbole, and Kommy Weldemariam. 2023. Scope 3 emission estimation using large language models. In *Proceedings of August 06-10, 2023 (Fragile Earth: AI for Climate Sustainability - from Wildfire Disaster Management to Public Health and Beyond).* ACM, New York, NY, USA, 8 pages. https://doi.org/10.1145/nnnnnnn.nnnnnnn

## 1 INTRODUCTION

The United Nations (UN) Sustainable Development Goals (SDGs) provide a framework for achieving a better and more sustainable future for all, with 17 goals and 169 targets adopted by all UN member states in 2015. Among these goals, SDG 13 focuses on combating climate change and its impacts, with a primary target of integrating

---

*Both authors contributed equally to this research.

**Unpublished working draft. Not for distribution.**

climate change measures into national policies, strategies, and planning [15]. Unfortunately, the world remains off track in meeting the Paris Agreement's target of limiting the temperature rise to 1.5°C above pre-industrial levels and reaching net-zero emissions by 2050 [14], with a projected temperature rise of around 2.7°C above pre-industrial levels by 2100 [22]. To achieve these targets, it is critical to engage non-state actors like enterprises, who have pledged to reduce their GHG emissions, and have significant potential to drive more ambitious actions towards climate targets than governments [9]. However, a lack of high-quality data and insights about an enterprise's operational performance can create barriers to reducing emissions. AI assisted tools can play a vital role in addressing these concerns and enabling an enterprise's decarbonization journey [24].

The GHG Protocol Corporate Standard [18] provides a systematic framework for measuring an enterprise's GHG emissions, classifying them into three scopes. Scope 1 (S1) emissions are direct emissions from owned or controlled sources, while Scope 2 (S2) emissions are indirect emissions from the generation of purchased energy. Scope 3 (S3) emissions are all indirect emissions that occur in the value chain of the reporting company, including both upstream and downstream emissions. To provide companies with a more detailed picture of their S3 emissions, they are further divided into 15 distinct reporting categories, such as Purchased Goods and Services, Capital Goods, and Business Travel. However, since S3 emissions occur outside of an enterprise's direct control or ownership, tracking and measuring them can be extremely challenging.

Despite the importance of S3 emissions in reducing GHG emissions, most enterprises have focused on reducing their S1 and S2 emissions, leaving S3 emissions largely unabated. However, for many enterprises, S3 emissions make up the majority of their emission inventories. A report by [9] found that reducing global S3 emissions by an additional 54% from current baseline scenarios could help achieve a climate reduction target of limiting temperature rise to 1.75° C above pre-industrial levels by 2100, instead of the projected 2.7° C. Since S3 emissions often overlap with other enterprises' emission inventories, reducing S3 emissions not only has the potential to mitigate the impact of climate change but can also provide significant business benefits [5].

Foundation models are a recent breakthrough in the field of artificial intelligence, comprising large models trained on vast datasets using self-supervised learning techniques. Once trained, these models can be fine-tuned for various downstream tasks, yielding superior performance compared to traditional machine learning and deep learning models. While foundation models have demonstrated remarkable success in numerous domains, their application in the context of climate and sustainability has yet to be fully explored. The potential of foundation models to aid in reducing greenhouse

gas emissions and achieving sustainable development goals calls for further investigation.

Prior research has started to explore the potential of foundation models in the domain of climate and sustainability. For instance, [17] used a domain-specific language model pre-trained on business and financial news data to identify Environmental, Social and Governance (ESG) topics. [11] employed a custom transformer-based NLP model to extract climate-relevant passages from financial documents. Similarly, [3] investigated the application of pre-trained transformers based on the BERT architecture to classify sentences in a dataset of climate action plans submitted to the United Nations following the 2015 Paris Agreement. In addition, [7] proposed the development of a benchmark comprising several downstream tasks related to climate change to encourage the creation of foundation models for Earth monitoring. However, none of these studies have focused on utilizing foundation models for estimating S3 emissions.

In this paper, we proposed a *first-of-a-kind* framework for estimating Scope 3 emission leveraging large language models by classifying transaction/ledger description of purchase good and services into US EPA EEIO commodity classes. We conduct extensive experiments with multiple state of the art large language models (roberta-base, bert-base-uncased and distilroberta-base-climate-f) as well as non-foundation classical approaches (TF-IDF and Word2Vec) and investigate the potential of foundation models for estimating S3 emissions using financial transaction records as a proxy for goods and services. We compare the performance of conventional models with foundation models for commodity recognition and mapping of financial transaction records into United States Environmentally-extended input-output (US EEIO) commodity classes. We also present a case study based on a sample enterprise financial ledger transaction data to demonstrate the proposed framework.

## 2 BACKGROUND

### 2.1 Scope 3 challenges

Scope 3 emissions along the value chain account for around 80% of a company's greenhouse gas impacts, which makes it important for companies to account for their Scope 3 emissions along with their Scope 1 and 2 emissions in order to develop a full GHG emissions inventory. This is also required to enable companies to understand their full value chain emissions and to focus their efforts on the greatest GHG reduction opportunities. Most of the largest companies in the world now account and report the emissions from their direct operations (Scopes 1 and 2). However, merely few companies are currently reporting their Scope 3 emissions for limited categories. The primary reason behind this contrast is the immense complexity associated with tracking and accounting of Scope 3 emissions. Some of the primary challenges faced by enterprises today are as follows:

- **Daunting data collection process:** Organizations struggle to collect relevant and sufficiently granular primary data and to manage the amount of data needed to calculate Scope 3 emissions.
- **Lack of Cooperation:** In order to successfully manage their Scope 3 emissions, enterprises need a certain extent of collaboration with third parties like upstream suppliers,

lessors/lessees, employees or customers. Enterprises today face a lack of cooperation along the value chain, which hinders them from successfully managing Scope 3 emissions.

- **Lack of Transparency:** The lack of transparency regarding the relevance of Scope 3 emissions is another major challenge for many companies. They may also lack transparency into their upstream and downstream partners' data collection and calculation processes, which reduces the overall quality of information exchanged.
- **Large Set of Stakeholders:** Scope 3 involves a large set of stakeholders across processes, including the upstream and downstream value chain. This makes the task of data collection more complex and also makes it challenging for enterprises to track and measure their Scope 3 emissions.
- **Lack of Personnel Resources:** The calculation of Scope 3 emissions requires personnel, resources, expertise, and data management and quality processes. Enterprises need a good management and leadership support in order to bring alignment which in itself is not a simple task.

### 2.2 Scope 3 calculation methodologies

The GHG Protocol Corporate Standard [18] recommends the use of Life Cycle Assessment (LCA) approach [23] or environmentally Extended Input-Output Analysis (EIOA) method [13] for Scope 3 reporting and estimation. The LCA approach evaluates the environmental footprints of a product or activity by considering the entire life-cycle of the processes involved starting from raw material extraction to the disposal. It requires the detailed knowledge about the activities and services involved in the supply chain and its associated emission factors. But the large companies have tens of thousand of products and services, and it is very complex and impractical to collect the detailed physical activity data across the supply chain. Alternatively, the most widely adopted method is environmentally Extended Input-Output Analysis (EIOA) method. It is based on the economic model called Input-Output Analysis (IOA) proposed by [8], which captures the inter and intra-trade relations between various sectors of economy across different countries. This economic model was extended to include the environmental impact called environmentally Extended Input-Output model, which estimates the amount of carbon footprints for delivering the one-dollar output by the sectors or products. It only requires to collect the value-based (monetary values) activity data of the company. It mainly consists of purchasing data and it is broadly classified into few product group or economic sectors. The emission factors for every dollar spend on each group of products or economic sectors can be determined from the EEIO framework [20].

Apart from these activity-data based and spend-based approaches, few research works have also focused on using machine learning techniques for estimating Scope 3 emissions. [21] built ML regressor models to predict Scope 3 emissions using widely available financial statement variables, Scope 1&2 emissions, and industrial classifications. [16] explored the quality of Scope 3 emission data in terms of divergence and composition, and the performance of machine-learning models in predicting Scope 3 emissions, using emission datasets from different data providers.

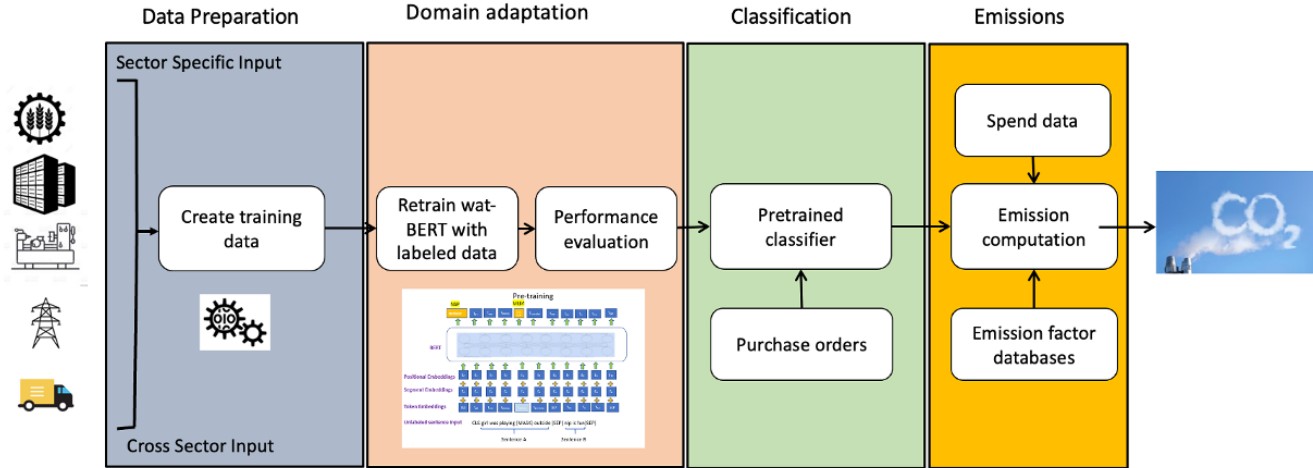

**Figure 1: Framework for estimating Scope3 using large language model**

## 2.3 USEEIO dataset

US Environmentally-Extended Input-Output (USEEIO) is a life cycle assessment (LCA) model that tracks the economic and environmental flows of goods and services in the United States. It provides comprehensive dataset and methodology that combines economic input-output analysis with environmental information to quantify environmental impacts associated with economic activities. USEEIO is a free and open-source dataset that is available for download from the US Environmental Protection Agency website.

USEEIO classifies commodities into over 380 categories of categories of goods and services that share similar environmental characteristics, which are called commodity classes. USEEIO provides emission factors for for these commodity classes, which are are to estimate environmental impact from spend data. These commodity classes are designed to align with the North American Industry Classification System (NAICS) and the Bureau of Economic Analysis (BEA) codes, where BEA codes aggregate multiple NAICS codes into higher level of industry descriptions. This alignment ensures consistency and facilitates the integration of USEEIO with existing economic and environmental datasets. Apart from the detailed commodity classes, USEEIO also provides emission factors for 66 summary commodity classes.

## 2.4 Why Foundation Model

In last couple of years, big supply chain players are trying their best to collect primary data from their suppliers through multiple mechanism e.g. supplier selections, financial incentives, etc. Unfortunately, there is no notable progress due to the challenges mentioned in Section 2.1. Therefore, it is imperative to use proxy data such as financial transactions for Scope 3 calculation. Spend data is easily available in any organization and is a good proxy of quantity of goods/services purchased and embedded emission.

While spend based Scope 3 emission estimation has an opportunity to address this complex problem, there are some challenges as follows:

- **Commodity recognition:** Purchased products and services are described in natural languages in various forms. Commodity recognition from purchase orders/ledger entry is extremely hard.
- **Commodity mapping:** There are millions of products and services. Spend based emission factor is not available for individual product or service categories. Organizations like US Environmental Protection Agenecy (EPA) or Organization for Economic Cooperation and Development (OECD) provide spend based emission factors for limited product/service categories. Manual mapping of the commodity/service to product/service category is extremely hard if not impossible.

Recent advances in deep learning based foundation models for natural language processing (NLP) have proven to outperform conventional machine learning models across a broad range of NLP classification tasks when availability of labeled data is insufficient or limited. Leveraging large pre-trained NLP models with domain adaptation with limited data has an immense potential to solve the Scope3 problem in climate and sustainability.

## 3 PROPOSED METHOD

We propose a first-of-a-kind AI based commodity name extraction and mapping service for scope3 emission estimation leveraging large language models. The framework is designed to process any Enterprise ledger transaction record written in natural languages to EEIO product/service category for estimating Scope 3 emissions. Figure 1 presents the proposed framework for estimating scope 3 emission using large language model. The framework consists of 4 modules i.e. data preparation, domain adaptation, classification, and emission computation. In the data preparation module, we thoughtfully created around 18000 examples of expenses (250-300 samples per class) for the 66 unique US EPA EEIO broad summary commodity classes. The samples are generated as per the International standard industrial classification of all economic activities (ISIC) with adequate representation from each sub-classes. In the subsequent module, we have chosen pre-trained foundation models

**Table 1: Values of hyperparameter varied during experimentation**

| Hyperparameter | Values |
|---|---|
| max_length | 64, 128, 256, 512 |
| Learning rate | 5e-5, 5e-6 |

(e.g. BERT, RoBERTa, ClimateBERT) and fine-tuned the foundation models for 66 commodity classes using the labeled samples with 70:20:10 train-validation-test split. Once the model is adapted, we evaluate its performance with reasonable test samples. We identified the classes with low performance and further fine-tuned the model with additional training samples for those classes. Once the domain adaptation is completed, the fine-tuned FM is deployed for inferencing on unknown data set. Now the fine-tuned model can take any ledger description written in natural language as input and provide the best matched industry/commodity class. Once, the commodity mapping is completed, the frame work can compute Scope 3 emission for each purchase line item using given EEIO emission factors.

This framework will help to avoid the humongous manual efforts and will accelerate the Scope 3 estimation at Enterprise scale. As financial transactions are easily accessible, the proposed method could make the Scope 3 calculation way more affordable at enterprise scale.

## 4 EXPERIMENT

We experiment with different training strategies including our proposed approach for the task of commodity classification in order to evaluate and compare their performance on the given problem.

## 4.1 Using Zero-shot classification

We tried zero-shot learning to evaluate if we can perform classification of transaction data to EEIO commodity classes, without using any domain specific training data.

*4.1.1 Using Semantic Text similarity.* [2] proposed carbon fooptprint of household products using Zero-shot semantic text similarity. This technique aims to assign predefined labels to text without any explicit training examples. It leverages pre-trained language models and sentence transformers, which are deep learning architectures specifically designed to encode textual information into fixed-length vector representations. By employing these models,

the semantic relatedness between input text and commodity classes can be effectively measured by means of a semantic similarity score between the embeddings of input text and commodity classes.

We tried few experimental settings to test Semantic Text similarity approach for our task-

- Commodity class titles vs Commodity class descriptions: As mentioned above, this approach works by finding semantic similarity between the embeddings of input text and commodity classes. For computing the embedding of commodity classes we tried both the titles of the commodity classes as well as their descriptions of their NAICS codes provided in the US Bureau of Labor Statistics Website [1]. For commodity classes which were comprised of more than one NIACS code, we concatenated the descriptions of the individual NAICS codes to create the commodity class description. For example, a commodity class *Food and beverage and tobacco products* (Commodity code 311FT) in USEEIO is comprised of 2 NAICS classes - *Food Manufacturing* and *Beverage and Tobacco Product Manufacturing*, having NAICS codes 311 and 312 respectively.
- Sentence transformer models: We additionally experimented with different open source sentence-transformer models available in *sentence-transformers* python library [19] to compute embeddings for the input transaction text and commodity names/descriptions. We tried `all-mpnet-base-v2`, `all-MiniLM-L12-v2` and `all-MiniLM-L6-v2` models.

We used cosine similarity between the embeddings of input text and commodity classes as the measure of semantic similarity, for all the experimental settings mentioned above.

## 4.2 Supervised learning using classical model (SL)

We experimented with supervised learning approach using classical ML model. It required feature vectorization of the input text in order to perform model training. We tried TF-IDF (Term Frequency-Inverse Document Frequency) and Word2Vec for feature vectorization of the input transaction texts.

*4.2.1 TF-IDF.* TF-IDF represents the importance of a term in a document, taking into account both its frequency in the document (TF) and its rarity across the entire corpus (IDF) and is a widely used technique in Information Retrieval and Text Mining We first performed feature extraction and calculated the TF-IDF values for

---

[1] https://www.bls.gov

**Table 2: Results for Zero-shot classification. TS- Semantic Text Similarity, Title/Description - Whether title or description of commodity classes was used**

| Approach | Model | Title/Description | F1 Score (test) |
|---|---|---|---|
| TS | all-mpnet-base-v2 | Title | 27.9% |
| | | Description | 43.7% |
| | all-MiniLM-L6-v2 | Title | 27.5% |
| | | Description | 40.1% |
| | all-MiniLM-L12-v2 | Title | 27.5% |
| | | Description | 40.9% |

each term in the training corpus. Then we perform feature vectorization wherein each input transaction text is converted to a numerical feature vector based on the TF-IDF values. We used the TF-IDF feature vector for training ML classifier model.

*4.2.2 Word2Vec.* Word2Vec introduced in [12], is a popular algorithm for generating word embeddings, which are dense vector representations of words in a high-dimensional space. It learns to encode semantic relationships between words by training on a large corpus of text. We used Word2Vec to create feature representation of the input transaction data for model training. This was done by computing the average of word embeddings for all the words in a given transaction text.

We trained random forest classifier models for both TF-IDF and Word2Vec based feature vectors representation.

## 4.3 Supervised fine-tuning for Scope3 (SFT3)

We additionally experimented with fine-tuning encoder based Large Language Models (LLMs) for classifying transaction data into commodity classes. These models are pre-trained using masked language modeling (MLM) and next sentence prediction (NSP) objectives. We tried the following commonly used models available on Huggingface.

*bert-base-uncased.* The BERT (Bidirectional Encoder Representations from Transformers) is a popular pre-trained model developed by [4]. BERT-base-uncased is uncased version of BERT model which is trained on a massive amount of textual data from books, articles, and websites. The training process involves a transformer-based architecture, which enables BERT to capture bidirectional dependencies between words and context. BERT-base-uncased consists of 12 transformer layers, each with a hidden size of 768, and has 110 million parameters.

*roberta-base.* RoBERTa (Robustly Optimized BERT pretraining Approach) is a variant of the BERT (Bidirectional Encoder Representations from Transformers) model developed by [10]. Similar to BERT, the RoBERTa-base also utilises dynamic masking during pre-training where tokens are randomly masked out. However, RoBERTa-base extends this approach by training with larger batch sizes, more iterations, and data from a more diverse range of sources. It also utilises a larger vocabulary and removes the next sentence prediction objective used in BERT, focusing solely on the masked language modeling objective. It excels in understanding the context and semantics of text, and captures subtle relationships and fine-grained nuances in language. It consists of 12 transformer layers, each with a hidden size of 768, and has 125 million parameters.

*climatebert/distilroberta-base-climate-f.* The ClimateBERT language model is additionally pre-trained, on top of DistilRoBERTa model, on text corpus comprising climate-related research paper abstracts, corporate and general news and reports from companies [25]. It has been shown improve the performance of various climate related downstream tasks [6]. The model has 6 transformer layers, 768 dimension and 12 heads, totalizing 82M parameters (compared to 125M parameters for RoBERTa-base).

We used *transformer* [26] library from Huggingface which allows to easily load and fine-tune pre-trained models from Huggingface.

**Table 3: Results for Supervised learning using classical model (SL). TF-IDF: Model was trained on TF-IDF based feature vectors, Word2Vec: Model was trained on Word2Vec based feature vectors**

|  | F1 (test) |
| --- | --- |
| TF-IDF | 69% |
| Word2Vec | 72% |

**Table 4: Results for supervised fine-tuning ($\alpha$ = 5e-5)**

| Model | max_length | F1 Score (test) |
| --- | --- | --- |
| roberta-base | 64 | 86.17% |
|  | 128 | 86.8% |
|  | 256 | 87.2% |
|  | 512 | 86.4% |
| bert-base-uncased | 64 | 86.6% |
|  | 128 | 86.38% |
|  | 256 | 86.32% |
|  | 512 | 86.44% |
| distilroberta-base-climate-f | 64 | 85.48% |
|  | 128 | 85.4% |
|  | 256 | 84.5% |
|  | 512 | 85.1 % |

**Table 5: Results for lower learning rate ($\alpha$ = 5e-6)**

| Model | max_length | F1 (test) |
| --- | --- | --- |
| roberta-base | 512 | 87.19% |
| bert-base-uncased | 512 | 87.14% |
| distilroberta-base-climate-f | 512 | 85.20% |

We experimented with varying the *max_length* parameter which determines the maximum length of the input sequences that the model can handle. We also experimented with lowering the default learning rate that is provided in the library. Table 1 shows the values of different values of hyperparameters that were tried as part of the experimentations. We evaluated the results on the model checkpoint with best validation loss by setting *load_best_model_at_end* argument to be *True*.

## 5 RESULTS AND DISCUSSIONS

### 5.1 Evaluation method

We evaluate the approaches detailed in the previous section by using the test data comprising of 1859 samples of transaction texts. We use weighted F1 score as the metric to compare the performance of the different approaches.

### 5.2 Comparison of training methods

*Performance of zero-shot method.* Table 2 shows the results from zero-shot approaches. We observe that some useful insights can be drawn from the results. F1 score is in the range of 27-28% when using the title of commodity classes for computing the embeddings.

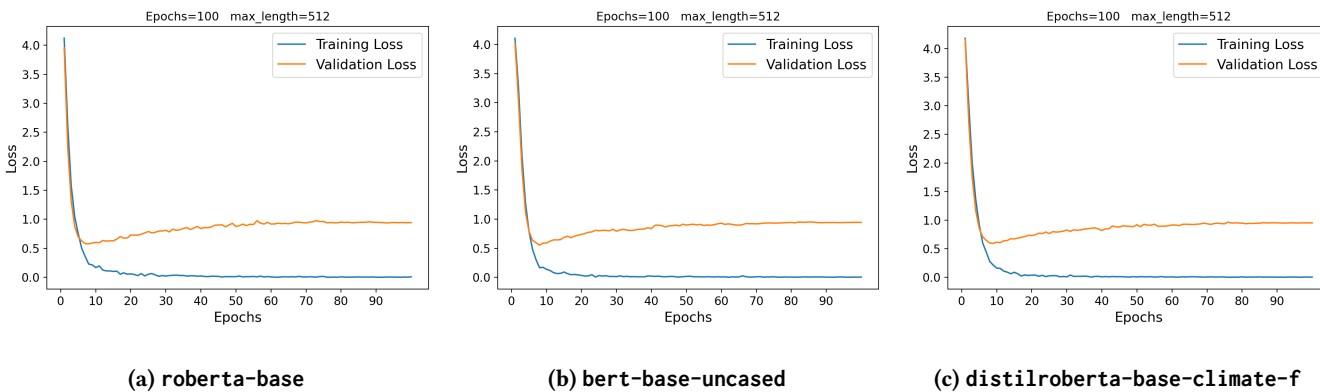

(a) `roberta-base`     (b) `bert-base-uncased`     (c) `distilroberta-base-climate-f`

**Figure 2: Learning curve showing variation in training and validation loss with epochs ($\alpha$=5e-5)**

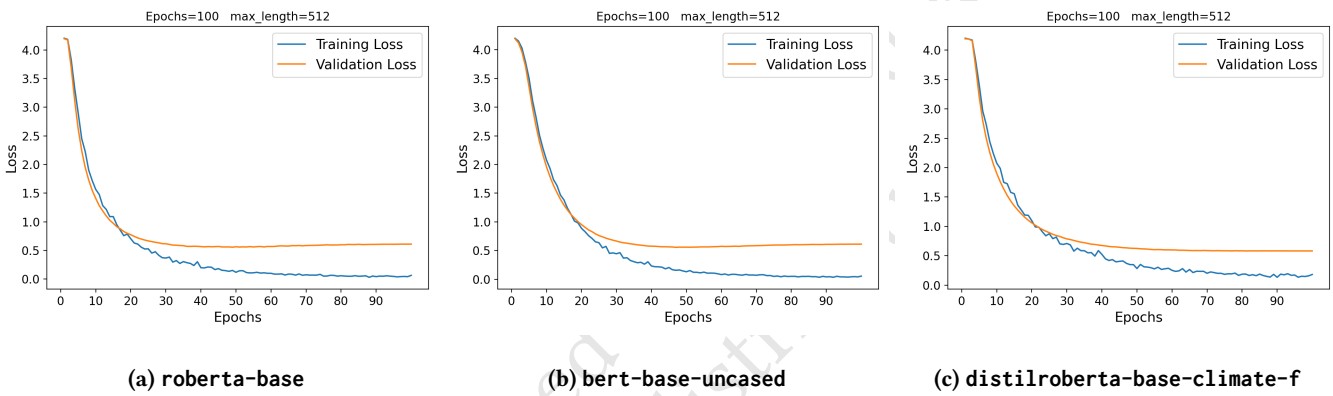

(a) `roberta-base`     (b) `bert-base-uncased`     (c) `distilroberta-base-climate-f`

**Figure 3: Learning curve showing variation training and validation loss with epochs ($\alpha$=5e-6)**

On the other hand, when the commodity description is used, the resulting F1 score is around 40-43 %. This makes sense, as descriptions provide more details about the commodity classes than their titles. Therefore embeddings computed using the descriptions are more meaningful and therefore result in a better performance. We also see that `all-mpnet-base-v2` performs better as compared to the other 2 sentence transformers. This is also in accordance with the details provided in documentation of the SBERT sentence transformers [2]. However, we see that the performance of the text similarity approach has an overall low performance with a low F1 score, highlighting the importance of a supervised approach for this task.

*Performance of Supervised Learning using classical models.* : Table 3 shows the performance of classical modeling using the TF-IDF and Word2Vec based vectorization of input. We see that Word2Vec has a better performance as compared to TF-IDF. This makes sense because Word2Vec captures the semantic relationships between words by representing them as dense vectors. This allows it to capture contextual and relational information that may not be captured by TF-IDF, which only considers term frequencies and inverse document frequencies.

We also see that overall this approach has a strong improvement over the zero shot classification approach. This can be due to the fact that Large Language models such as sentence transformers are pre-trained mainly on texts in the form of paragraphs and sentences. However in our problem, we see that the transaction data are often not in the form of complete sentences, but in the form of sentence fragments such as *Computer and Peripheral Equipment expense*, or *Farm crop packaging materials cost*. Hence LLMs are not able to generalize over these inputs in a zero-shot fashion and have a poor performance.

*Performance of Supervised Fine Tuning.* Table 4 shows the performance of `roberta-base`, `bert-base-uncased` and `distilroberta-base-climate-f` for different *max_length*, for a learning rate of 5e-5. We see that the performance of the models is not impacted by varying *max_length*. This is because the number of tokens in input texts are in a much smaller range. Hence, there was no truncation of the input text, which could have impacted the model's performance. We also observe that the performance of ClimateBERT model is lower as compared to the other 2 models, which could be due to the fact that it has fewer layers and parameters as compared to the other two models.

---

[2]www.sbert.net/docs/pretrained_models.html

**Figure 4: Commodity classes Spend and its Scope 3 emissions**

Fig 2 shows the variation in training and validation loss with epochs for *max_length* of 512, for the 3 LLM's used in the analysis. We observe that in all the 3 learning curves, the validation loss decreases sharply in the initial epochs and then starts to slightly increase after around the 10th epoch, while the training curve is almost constant. We repeated the training after lowering the learning rate from 5e-5 to 5e-6. Figure 3 shows the learning curve for the 3 models fine-tuned using the lower leaning rate. We observe that fine-tuning using the lower learning rate resulted in a smoother convergence and helped in achieving a better validation loss of the best checkpoint model. For example, in roberta-base, the validation loss dropped from 0.577 to 0.557, denoting a better fine-tuning. This also improved the F1 score on the test data for all the models, as can be seen from Table 5.

We also see that this approach outperforms the classical ML model using the TF-IDF and Word2Vec based vectorization of input. The drastic improvement in performance can be attributed to the fact that LLM's have already been pre-trained on a large corpus of texts. This helps the model in developing a semantic understanding and learning of contextualized word representations. Fine-tuning these models on the input data allows the model to adapt to the task specific patterns, features, and labels, refining their representations and predictions for more accurate classification.

## 5.3 Ablation Study

*Subset of training data.* We experimented with the supervised learning approaches mentioned in Sections 4.2 and 4.3 using 50% of the training and validation data. Table 6 shows the results from the different models. We see that there is a reduction in F1 score, when compared to the results from tables 3 and 4. This denotes that the reduced training data may not be sufficient for the model to learn effectively. With a smaller dataset, the model may struggle to capture the full range of patterns and variations present in the

**Table 6: Results for 50% smaller training data**

| Model | max_length | F1 (test) |
|---|---|---|
| TF-IDF | | 64% |
| Word2Vec | | 66% |
| roberta-base | 512 | 81.29% |
| bert-base-uncased | 512 | 82.18% |
| distilroberta-base-climate-f | 512 | 80.03% |

financial ledger descriptions, leading to decreased performance on test data. It also suggests that having an even larger training dataset may further help improve the model's performance.

## 5.4 Estimation and analysis of Scope 3 emission

Once the ledger descriptions are mapped into EEIO summary commodity classes, we use EEIO emission factors (emission per $ spend) provided in *Supply Chain GHG Emission Factors for US Commodities and Industries v1.1* [1] to compute carbon footprint related to the expenses.

Figure 4 shows the sample expense in USD and its associated emission distribution for various commodity classes for the considered use case. It is interesting to see that the correlation between the expense and the respective emissions across the commodity classes are not trivial and very counter-intuitive. Some commodity classes have high expenditure with lower emission and vice-versa. For example, the commodity classes like *Administration, Cleaning, General building, Professional services, etc.,* have relatively lower emissions when compared to their expenses. Whereas the commodity *Utilities* has relatively very high emission with respect to its expenditure. Similarly the emissions of few commodities like *Conveyance Systems, Waste, etc.,* are comparably varying with the

expenses. These detailed insights provides an opportunity for the enterprises to take appropriate climate actions for Scope3.

## 6 CONCLUSION

In this paper, we proposed a *first-of-a-kind* framework for estimating Scope 3 emission leveraging large language models by classifying transaction/ledger description of purchase good and services into US EPA EEIO commodity classes. We have conducted extensive experiments with multiple state of the art large language models (roberta-base, bert-base-uncased and distilroberta-base-climate-f) as well as non-foundation classical approaches (TF-IDF and Word2Vec ) and found that domain adapted foundation models are outperforming the classical approaches. We also see that supervised fine-tuned foundation model approach has a strong improvement over the zero shot classification approach. While there is not significant performance variation across the chosen foundation models, we see roberta-base performs slightly better than others for the considered use case. We further conducted experiments with hyperparameter tuning (learning rate and maximum sequence length) for the foundation models, and found that model performance is not sensitive to maximum sequence length but the lower learning rate provides better convergence characteristics. The experimental results show that the domain adapted foundation model performs as good as domain-expert classification, and even outperforms domain-expert in many instances. As financial transactions are easily accessible, the proposed method will certainly help to accelerate the Scope 3 calculation at an enterprise scale and will help to take appropriate climate actions to achieve SDG 13.

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
