# OpenReview forum: "Scope 3 emission estimation using large language models"
_KDD.org/2023/Workshop/Fragile_Earth — KDD 2023 Workshop Fragile Earth Submission_

### Official Review · Reviewer_Md7y · 2023-07-05
**review of "Scope 3 emission estimation using large language models"**

**Rating:** 7
**Confidence:** 2

**Review:**

Summary:
The paper presents a study on the use of large language models for estimating Scope 3 emissions. It explores different approaches including zero-shot classification using semantic text similarity, supervised learning using classical models, and supervised fine-tuning of models like RoBERTa and BERT. The paper also discusses the use of different feature vectorization techniques such as TF-IDF and Word2Vec.

Strengths
1)	The paper presents a novel application of AI for estimating Scope 3 emissions, which is a significant contribution to the field of environmental science and sustainability.
2)	The authors have provided a detailed comparison of the performance of different models and approaches, which can be valuable for readers and other researchers in the field.

Weaknesses
1)	The paper does not provide a clear explanation of how the models handle the variability and uncertainty inherent in estimating indirect emissions, which could be a significant limitation in the practical application of the proposed method.
2)	The paper does not provide a clear explanation of how the performance of the models was evaluated. More details about the evaluation process, such as the metrics used and the specific results obtained, would be beneficial.

Questions
1)	How does the model's performance compare with other existing methods for estimating Scope 3 emissions?
2)	How can the proposed method be integrated into a company's existing systems and processes for environmental reporting?

---

### Official Review · Reviewer_CRRb · 2023-07-10
**Review of "Scope 3 emission estimation using large language models"**

**Rating:** 7
**Confidence:** 4

**Review:**

This paper aims to address the estimation of supply chain emissions problem using large language models (LLM). The paper is interesting and provides a different perspective by using a more generic (LLM with domain adaptation) approach instead of a curated approach specific to the problem. Overall, the paper is interesting and I will be happy to see it presented at the venue.

However, I have a couple of questions going forward.

a. Scope 3 is meaningless for a person who is outside this domain. For example, I didn't know these details until I read them in the introduction. Is it possible to replace Scope 3 -> Supply Chain Emissions? By doing this change, it will be easier for interested individuals to search for the topic more easily as well.
b. Emissions problem may have its own statistical models, as well as estimation models that use features generated from the financial transactions. Section 2.2 already refers to those. However, the paper chose to compare the approach with other NLP approaches. Is there a reason for that?
c. Do you think 18000 samples is enough especially considering 66 different commodity classes? What could unblock collection of more samples?
d. Figure 3 is pretty much meaningless in terms of the additional insights it may provide. I would rather remove it and add a discussion regarding what can further improve the proposed model. And/or add a comparison with a non-NLP method.
e. Can you provide an intuitive explaination about why the results change in Table 4? They do not look statistically significantly different. Did you run the train/inference multiple times to reduce the bias caused by the model initialization?

---

### Official Review · Reviewer_vANU · 2023-07-11
**This is a good paper addressing an extremely topical and critical problem of carbon footprint estimation towards meeting sustainable development goals (SDG 13) from natural language reports using fine-tuned foundation models.**

**Rating:** 8
**Confidence:** 4

**Review:**

This paper addresses the need to estimate carbon foot prints of enterprise activities in supply chains and hence are indirect (referred to as "Scope 3" emissions), in order to properly incentivize enterprises to measure and reduce their carbon emissions across broad business and industry sectors. The authors tackle this problem using textual report data on business transaction data, using large language models to solve important sub-problems like commodity recognition and mapping. The paper is well-written, and articulates the importance of the problem, the challenges involved, and the approach and framework they propose for this problem.
The technical work is primarily that of using pre-trained large language models and fine-tuning them to customize to the problem domain. They conduct extensive experiments to demonstrate that the proposed approach out-performs classical machine learning based methods in those tasks. The paper also exhibits the overall, end-to-end framework inclusive of the data collection, data preparation and carbon emission computation steps, which will be important in paving the way towards real world deployment.
The work is promising and it can be expected that future deployment and utilization of the proopsed framework could have non-trivial impact on the Scope 3 emission estimation, and hence, towards the larger goal of climate change mitigation in general.

---

### Official Review · Reviewer_aeC3 · 2023-07-13
**Review for "Scope 3 emission estimation using large language models Download PDF KDD 2023 Workshop Fragile Earth"**

**Rating:** 7
**Confidence:** 2

**Review:**

Summary : Use case of NLP based foundation models has been presented where already trained models like BERT are fine tuned to analyze document and texts that capture Scope 3 (Supply chain) emission data. It has also been compared with other NLP models and results have been provided.

Strengths :
- Positive results are shown and the large language based models are compared with other NLP based methods.
- Sufficiently thorough analysis of the problem has been provided.
- Ablation study is given.

Weaknesses :
 - No limitations or future work has been provided.

Question : what is the underlying environmental cost of training such models? Is there any work or study or how much the method and deployment of LLM/Foundation models cost?

---

### Decision · Program_Chairs · 2023-07-19

**Decision:**

Accept (Oral)

**Comment:**

Congratulations!

We are pleased to inform you that your submission: Scope 3 emission estimation using large language models has been accepted to The KDD 2023 Workshop Fragile Earth: AI for Climate Sustainability - from Wildfire Disaster Management to Public Health and Beyond.

Camera ready deadline is ** July 24 AOE **.  Please log in to OpenReview and prepare your camera-ready version based on the reviews. Formatting rules are the same as for the initial submission and submissions must adhere to KDD 2023 guidelines available at https://authors.acm.org/proceedings/production-information/taps-production-workflow.

Again, congratulations on the acceptance of your paper!  We look forward to seeing you at the workshop on Aug 7, 2023.

The Fragile Earth Workshop Proceeding Chairs